# Rapid cost decrease of renewables and storage accelerates the decarbonization of China's power system

Gang He[1,2✉], Jiang Lin[2,3✉], Froylan Sifuentes[2,4], Xu Liu[2], Nikit Abhyankar[2] & Amol Phadke[2✉]

The costs for solar photovoltaics, wind, and battery storage have dropped markedly since 2010, however, many recent studies and reports around the world have not adequately captured such dramatic decrease. Those costs are projected to decline further in the near future, bringing new prospects for the widespread penetration of renewables and extensive power-sector decarbonization that previous policy discussions did not fully consider. Here we show if cost trends for renewables continue, 62% of China's electricity could come from non-fossil sources by 2030 at a cost that is 11% lower than achieved through a business-as-usual approach. Further, China's power sector could cut half of its 2015 carbon emissions at a cost about 6% lower compared to business-as-usual conditions.

[1] Department of Technology and Society, College of Engineering and Applied Sciences, Stony Brook University, Stony Brook, NY 11794, USA. [2] International Energy Analysis Department, Lawrence Berkeley National Laboratory, Berkeley, CA 94720, USA. [3] Department of Agricultural and Resources Economics, University of California, Berkeley, Berkeley, CA 94720, USA. [4] Institute of Energy Studies, Environmental Sciences Department, Western Washington University, Bellingham, WA 98225, USA. ✉email: gang.he@stonybrook.edu; j_lin@lbl.gov; aaphadke@lbl.gov

China's electricity system accounts for about half of the country's energy-related carbon dioxide ($CO_2$) emissions, which represent about 14% of total global energy-related $CO_2$ emissions[1]. Decarbonizing China's electrical system therefore is essential to the decarbonization of energy systems not only in China but also globally. Further, given electricity's increasing role in China's energy use, a low-carbon electrical system is key to reducing $CO_2$ emissions from other economic sectors such as transport, industry, and buildings.

Under the Paris Agreement, China committed to peak its $CO_2$ emissions and to supply 20% of its energy demand using non-fossil sources by 2030. Such targets, however, are unlikely to limit the worldwide temperature increase to 2 or 1.5 degrees above pre-industrial levels[2]. Various studies have outlined strategies for China to attain a high degree of non-emitting generation by 2050[3–6]. Many recent studies and reports around the world have not adequately captured the dramatic decrease in costs of renewable energy and storage, however. For example, the World Energy Outlook produced by the International Energy Agency and the International Energy Outlook developed by the U.S. Energy Information Administration have under-estimated the development of renewables[7–9].

Incorporating the new downward trend in costs of renewable energy into models of the power sector is both relevant to modeling efforts and required for developing appropriate policies. The analysis described herein aims to incorporate recent trends in renewable and storage costs so as to explore more ambitious pathways to decarbonizing China's power system by about 2030 and to offer insights on how those recent trends can reshape the power system. The costs of solar photovoltaics (PV), wind, and battery storage have decreased rapidly. The global weighted-average levelized cost of electricity (LCOE) of utility-scale solar PV, onshore wind, and battery storage has fallen by 77%, 35%, and 85% between 2010 and 2018, respectively[10–13]. Those cost trends bring new possibilities for widespread penetration of renewable energy sources and comprehensive power-sector decarbonization that were not foreseen in previous policy discussions.

We focus on the following questions in this study: how would China's power system change given the rapid decrease in costs of renewables and storage under more stringent $CO_2$ emissions targets? What are the costs to achieve those changes in China's power system? How would those changes affect China's regional pattern of power development and transmission? By addressing those questions, this paper is the first effort to reveal the implications of cost decrease on power systems and new perspectives on clean power transition that are not visioned in the existing literature.

We updated the SWITCH-China model[14] and developed four scenarios for 2030 to simulate and understand the effects of the rapid decrease in renewable energy costs. The scenarios are: First, business as usual scenario (BAU), which assumes the continuation of current policies and moderate cost decreases in future renewable costs. Second, low-cost renewables scenario (R), which assumes the rapid decrease in costs for renewables and storage will continue. Third, carbon constraints scenario (C50), which has a carbon cap of 50% lower than the 2015 level in 2030 on top of the R scenario. Fourth, deep carbon constraints scenario (C80), which further constrain the carbon emissions from the power sector to be 80% lower than the 2015 level by 2030.

Our modeling analysis shows if cost trends for renewables continue, 62% of China's electricity could come from non-fossil sources by 2030 at a cost that is 11% lower than achieved through a business-as-usual approach. Further, China's power sector could cut half of its 2015 carbon emissions at a cost about 6% lower compared to business-as-usual conditions. An 80% reduction in 2015 carbon emissions is technically feasible as early as 2030, but requires about a 21% higher cost than the business-as-usual approach, for a \$21/t$CO_2$ cost of conserved carbon.

## Results

**Mix of generation capacities and power generation.** As expected, rapid decreases in the costs of renewable energy sources lead to the larger installation of wind and solar capacity. By 2030, the low-cost renewables (R) scenario, compared with the BAU scenario, would lead to an increase in wind capacity from 660 to 850 GW and in solar capacity from 350 to 1260 GW. The need for power sector generators to incorporate flexibility in utilizing resources would result in increasing storage capacity from 34 to 290 GW to support the integration of variable renewable resources. The need for natural gas capacity would decrease from 300 to 170 GW, replaced by increasing renewable capacities and storage capacities. Coal capacity would diminish from 750 to 700 GW (Fig. 1), about a 7% reduction.

Under the carbon constraints (C50) scenario, coal capacity would decrease further to 520 GW by 2030, almost a 1/3 reduction compared with the BAU scenario. The deep carbon constraints (C80) scenario would phase out coal further to about 200 GW, only 4% of total capacity. The decrease in coal use would be offset primarily by renewables: 1920 GW of solar and 2000 GW of wind.

Under R scenario, coal-based generation would decrease from 4900 TWh in the BAU scenario to 3000 TWh by 2030, a 30% reduction. Wind and solar production could provide 39% of electricity need, with battery storage and natural gas supplementing the increasing wind and solar supplies. The total share of non-fossil generation could reach 62% in 2030. The C50 scenario would cause coal generation to decline further to 2400 TWh (less than half the amount generated under the BAU scenario), while the share of non-fossil generation would increase to 77% in 2030. The C80 scenario would reduce coal generation to about 960 TWh, or to about 10% of total power generation, while the share of non-fossil generation would approach 90% in 2030 (Fig. 2).

Relying on variable wind and solar resources for electricity could pose challenges to system operations. On days with abundant wind and solar resources, upto 300 GW of storage would be needed to balance the power system under the R scenario. On days that provide minimal solar and wind power, storage would be inadequate to make up for the shortage; natural gas generation could fill the gap in order to satisfy peak load requirements. Fig. 3 shows that dispatch sources to meet demands could be operationally manageable with the addition of electricity from battery storage and natural gas, represented in both the R scenario and the C50 scenario.

**Power costs and carbon emissions.** A low-cost renewables (R) scenario could reduce carbon emissions significantly, from 3980 Mt$CO_2$ under the BAU scenario (5% above the 2015 level) to 2970 Mt$CO_2$ by 2030 (22% below the 2015 level), see Fig. 4a. Given the remarkable and ongoing reductions in the cost of renewable power, this 30% reduction in carbon emissions could be achieved for a lower cost of power under the R scenario than under the BAU scenario, see Fig. 4b. Power costs would decrease from 73.52 \$/MWh under the BAU scenario to 65.08 \$/MWh under the R scenario, an 11% reduction. Under the carbon constraint (C50) scenario, carbon emissions in 2030 would be half of those of 2015, on a trajectory to achieve an 80% reduction by 2050. The cost of power under the C50 scenario is calculated to be 69.47 \$/MWh, only 7% higher than under the R scenario, but still 6% lower than under the BAU. In the deep carbon constraint (C80) scenario, the cost of power would increase to 89.08 \$/MWh, and 21% higher than under the BAU scenario. The cost

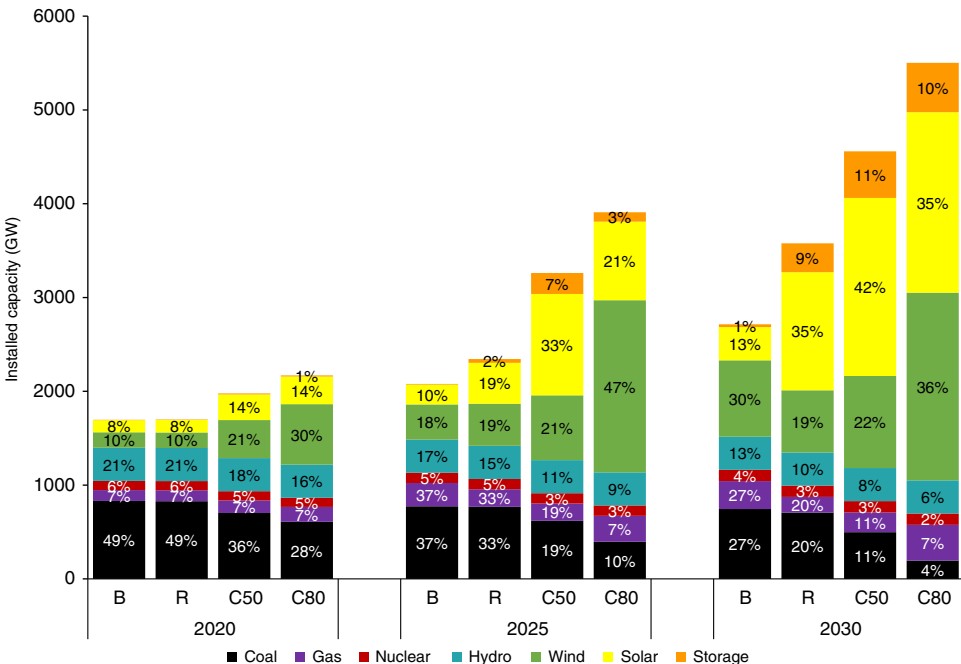

**Fig. 1 National capacity mix for four scenarios in 2020, 2025, and 2030.** The scale of the bar chart are the installed capacity by technologies, and the data labels show the share of each technology in total capacity. Source data are provided as a Source Data file.

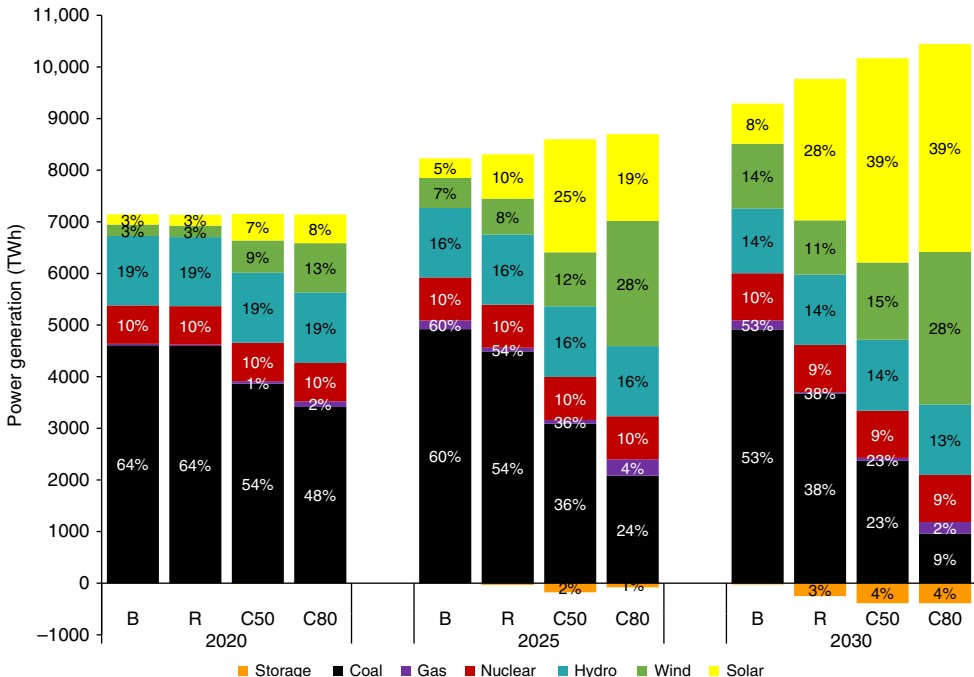

**Fig. 2 National power-generating mix for four scenarios in 2020, 2025, and 2030.** The scale of the bar chart are the generation by technologies, and the data labels show the share of generation by each technology in total generation. Source data are provided as a Source Data file.

of conserved $CO_2$ would be -$36/tCO_2$, -$9/tCO_2$, and $21/tCO_2$ under the R scenario, C50 scenario, and C80 scenario, respectively. China has already initiated a national cap-and-trade program limiting the carbon emissions from the power sector with a carbon price ranging from 20 RMB/$tCO_2$ (~$3/tCO_2$) to 100 RMB/$tCO_2$ (~$14.5/tCO_2$).

**Changing Investment Mix.** A low-cost renewables (R) scenario would shift the cost structure of the power system from a fuel intensive system to a more capital investment driven system, see

Fig. 5. The fuel cost of coal plants would decrease from about $100 billion in the BAU scenario to about $65 billion in the R scenario. New capital investment of solar, wind, and storage capacity in the R scenario is only slightly higher than the BAU scenario contribute to the lower cost of renewables and storage, from $55 billion in the BAU scenario to about $65 billion in the R scenario. The overall power system cost in the R scenario is $280 billion, 11% lower than that in the BAU scenario, $310 billion. Total costs under C50 and C80 are $285 billion and $390 billion, respectively in 2030.

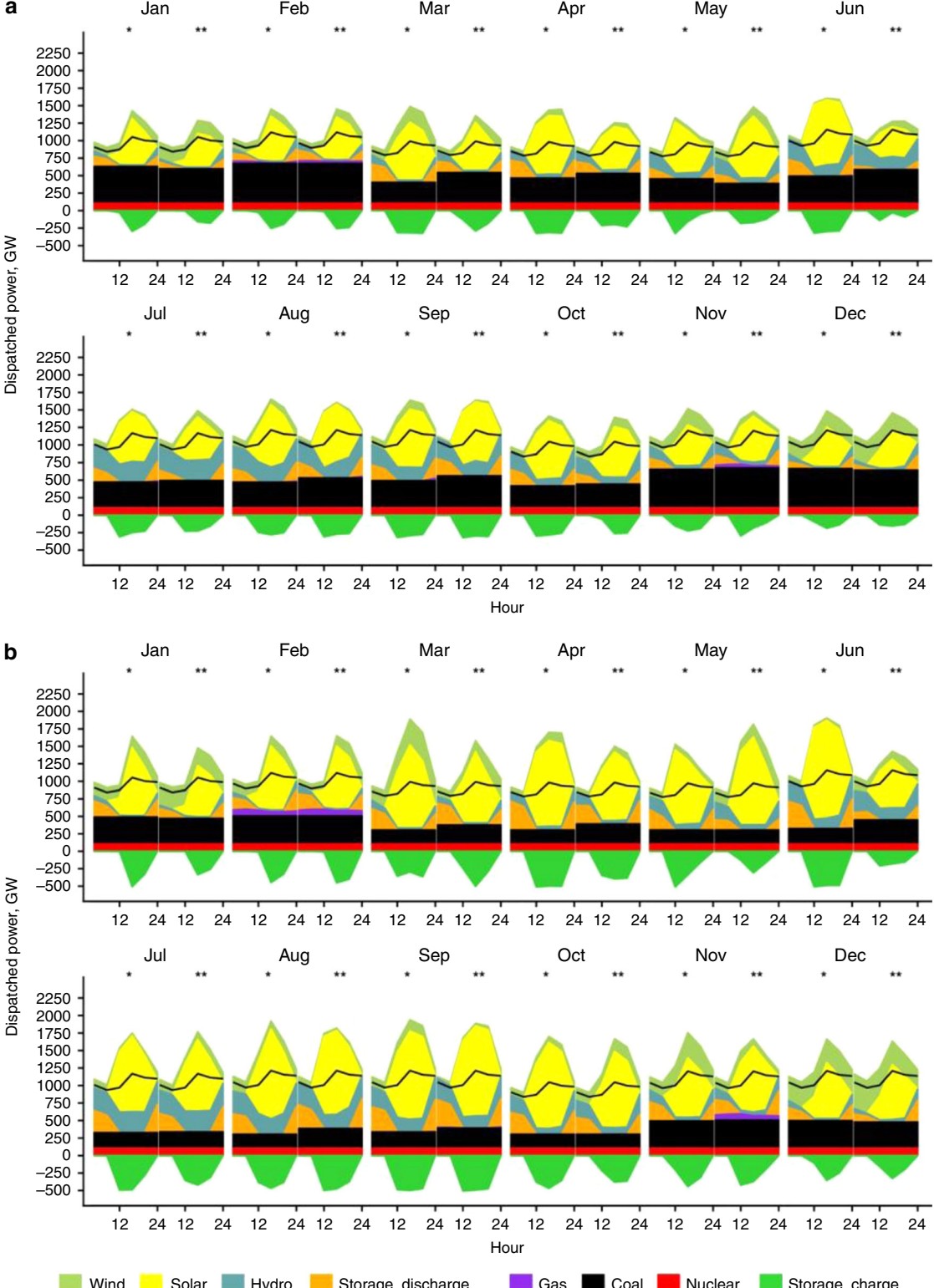

**Fig. 3 Hourly dispatch sources in 2030 under the R and C50 scenarios.** Two days, a normal day (●) and a peak day (●●) are selected in each month to represent the month. Black solid line is the system load. **a** Hourly dispatch in the R scenario. **b** Hourly dispatch in the C50 scenario. Source data are provided as a Source Data file.

**Regional disparities and needs.** Mapping the mix of resource capacity and required new transmission under the R scenario reveals regional disparities in the development of renewable energy sources (Fig. 6). First, solar capacities are concentrated in the northwest—in the provinces of Inner Mongolia, Qinghai and Shaanxi. Each of those areas has more than 100 GW of solar capacity. Wind capacities are more evenly distributed along the northwest, northeast, and eastern coastal provinces. Xinjiang, Heilongjiang, Shaanxi, Guangxi, Jilin, and Shanxi provinces have the greatest number of wind installations; each has more than 30 GW

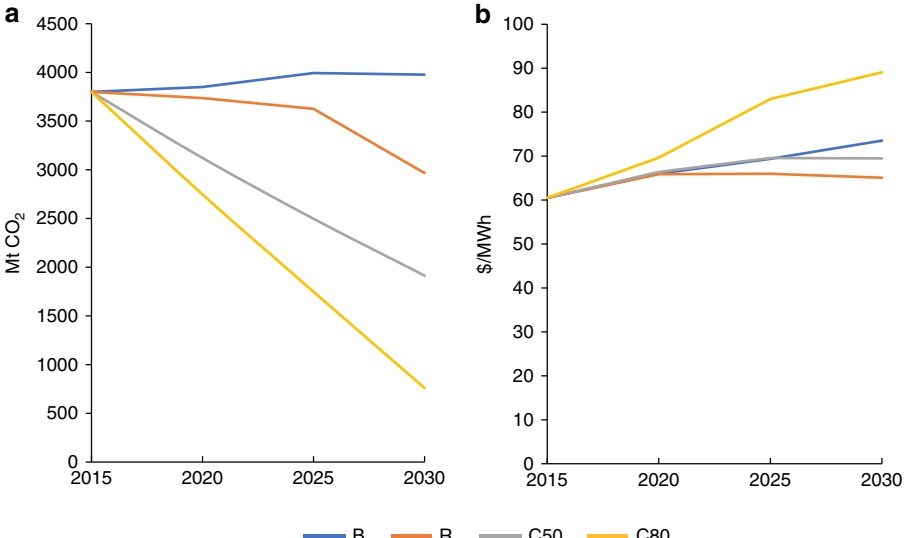

**Fig. 4 Carbon emissions and power costs to 2030 under four scenarios. a** Carbon emissions and **b** Power costs are shown in the business as usual scenario (B), the low-cost renewables scenario (R), the carbon constraints scenario (C50), and the deep carbon constraints scenario (C80), respectively. Power costs in the R scenario and the C50 scenario are 11% and 6% lower than that of the BAU scenario in 2030, respectively. Source data are provided as a Source Data file.

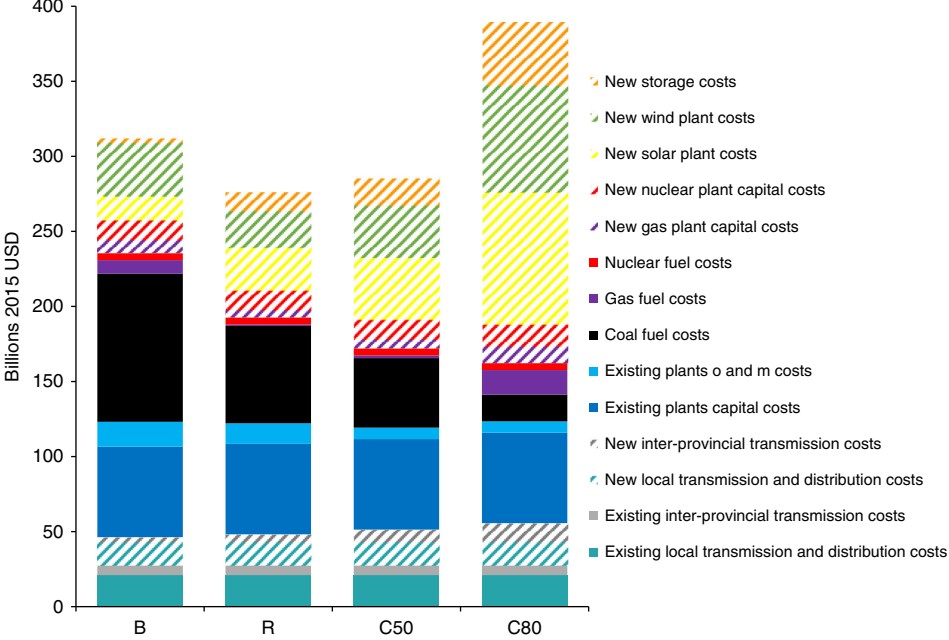

**Fig. 5 Distribution and costs of power sources under four scenarios in 2030.** The costs are categorized into existing and new capacity and transmission. Fossil fuel technologies and nuclear have fuel costs. Source data are provided as a Source Data file.

of wind capacity. Bringing the power generated from renewable sources to the areas of demand requires extensive transmission infrastructure. The focus for new transmission capacities are the three metropolitan areas of Jing-Jin-Ji, the Yangtze Delta, and the Pearl River Delta. New transmission infrastructure is needed to bring wind and solar energy from the northwest (Qinghai, Gansu, Inner Mongolia, and Shaanxi) to the central and eastern China grids; for example, from Inner Mongolia to Hebei, Beijing, and Tianjin; from Yunnan to Guangxi and Guangdong; from Anhui and Jiangsu to Zhejiang and Shanghai. The necessary transmission capacity could be as great as 35 GW, which would double the current maximum cross-provincial transmission capacity.

As shown in Fig. 7, regional disparities in demand, as well as resource availability, from hydro, solar, and wind, lead to

different generation profiles for the BAU scenario. Across almost all regions, for the BAU scenario, generation closely matches the regional demand. The only exception is the Northwest grid which, even under the BAU scenario, is expected to export electricity to other regional grids. Under the R scenario, solar and wind resources rich regions increase their electricity generation dramatically, while regions with less solar and wind availability see a decrease in their overall generation. Decreases in generation across the Central, Eastern, and Southern grids, come mostly from decreased electricity generation from coal in these regions. This trend continues as the scenarios impose increasingly stringent decarbonization goals across the national grid, as shown in the third and fourth columns in each regional graph.

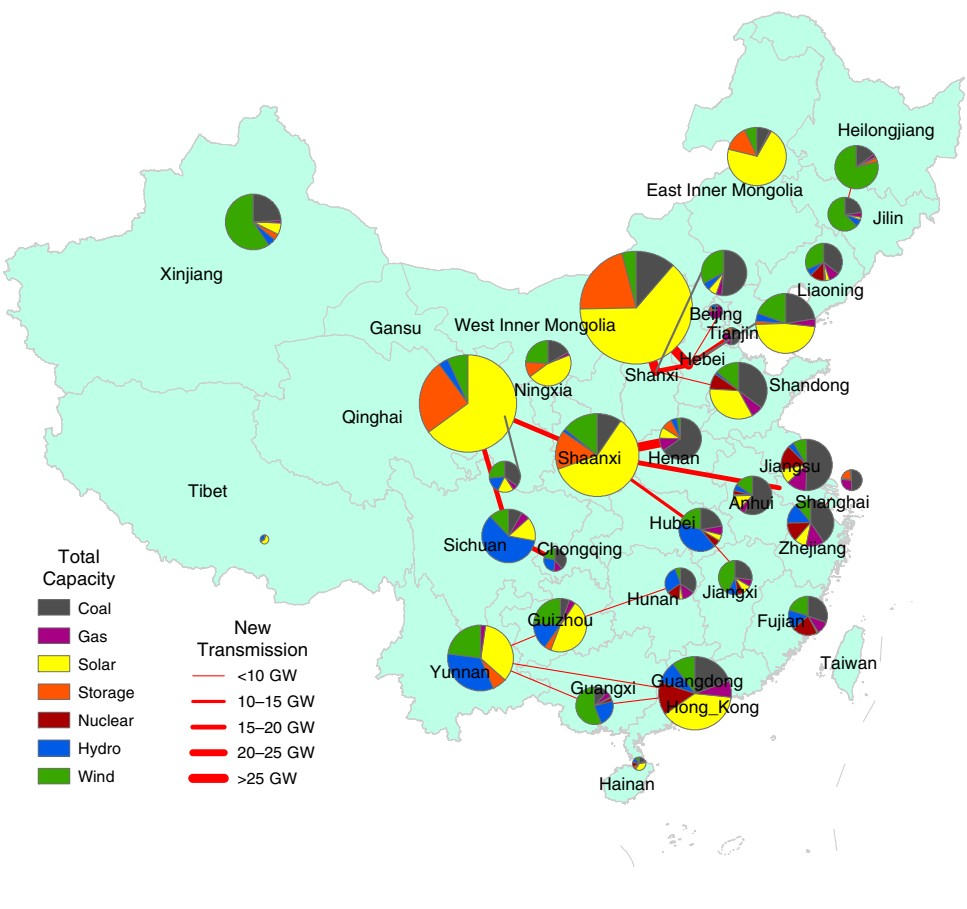

**Fig. 6 Provincial total capacity mix and new transmission lines required by 2030 under the R scenario.** The pie charts shows the total power capacity mix in each province, and the red lines show the new interprovincial transmission lines to bring electricity from resource centers to demand centers. Source data are provided as a Source Data file.

Under the R scenario, the Northwest grid is a net exporter of electricity to the Central, Northern, and Eastern grids. In particular, in 2030, under this scenario, the Northwest grid is expected to export 672, 287, and 90 TWh to the Central, Northern, and Eastern grids, respectively. On the other extreme, under the same scenario in 2030, the Eastern grid imports 287, 125, 111, 57, and 22 TWh from the Northwest, North, Central, South and Northeast grids. As outlined in this research, and as decarbonization priorities increase in the C50 and C80 scenarios, the total electricity generation in the Eastern grid further decreases and becoming increasingly import-dependent to meet its demand. Future studies might consider studying the impact of a decrease in costs for offshore wind, and demand response technologies on the Eastern grid's reliance on imports to meet its demand under more stringent decarbonization goals by 2030 and beyond.

Under our current assumptions, we can observe that with the assumed decrease in costs for solar, wind and storage technologies the Northwest region emerges as a national supplier of carbon neutral electricity even as that choice requires increases in transmission capacity across the Northwest and all other regions. Although not shown by arrows in Fig. 7, one can infer that this trend is further exacerbated by more stringent carbon reduction goals across the national grid. In particular, we can see that under the C80 scenario, the Northwest grid generation exceeds its own demand by over 300%, while the Eastern grid produces only about 50% of its total electricity demand.

**Sensitivity analysis and uncertainties**. The power sector is a dynamic, evolving system affected by costs, demands, and other factors. We conducted sensitivity analyses on two key assumptions: the capital costs of renewables (solar, wind, and storage), and future electricity demand. Changes in both resource capacity and generation respond to changes in demand and costs. We consider two sensitivity scenarios: D + 20% assumes that demand increases linearly 20% until 2030; C + 20% assumes that the capital costs of solar, wind, and storage are 20% higher than under the R scenario. Under the D + 20% scenario, by 2030 the capacities of solar and wind installations increase to 1890 GW and 1,040 GW, respectively, whereas under the C + 20% scenario, by 2030 the capacities of solar and wind installations decrease to 980 GW and 650 GW, respectively (Fig. 8a). The generation mix in the sensitivity scenarios follows a very similar pattern as in the capacity mix (Fig. 8b).

The large-scale decarbonization of the power sector requires that several processes take place simultaneously. First, both the resource capacity and transmission infrastructure must be scaled up quickly. Second, the investment needed for the infrastructure transformation must be acquired and dedicated. Third, social and economic equity must be addressed during the transition to lower carbon power systems. Any or all of those processes could encounter issues with the current physical framework and face obstruction from current stakeholders.

There is also uncertainty to deploy large-scale of storage capacity to integrate the renewables. Our results show in the R

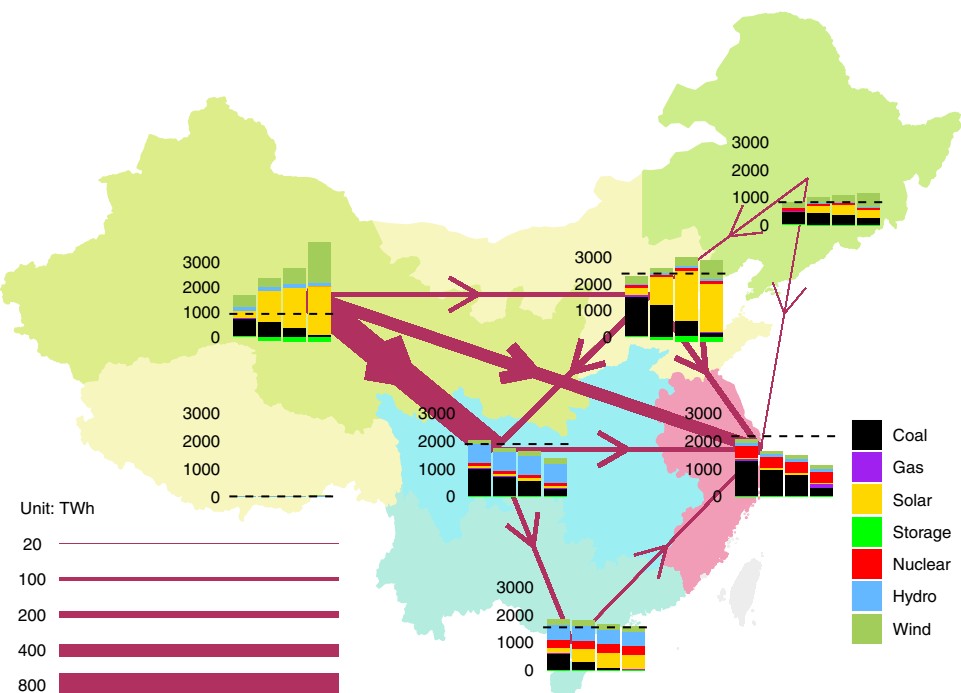

**Fig. 7 Regional generation, demand, and interregional transmission map for the R scenario in 2030.** The different grids are shaded in different colors based on the dominating energy source as the region decarbonizes. For example, the Northeastern grid is dominated by high wind energy penetration and is therefore shaded green and the Central grid is dominated by hydro electricity generation and is therefore shaded blue. Each region shows a graph with four bars representing the generation for the four different scenarios in order of increased carbon reduction (from left to right: BAU, R, C50, and C80, respectively). The dotted line across all bars in each set of generation graphs represents the yearly demand in 2030 in each region, which stays constant across the four scenarios. The magenta arrows point in the direction of the transmission flow between two regions. Source data are provided as a Source Data file.

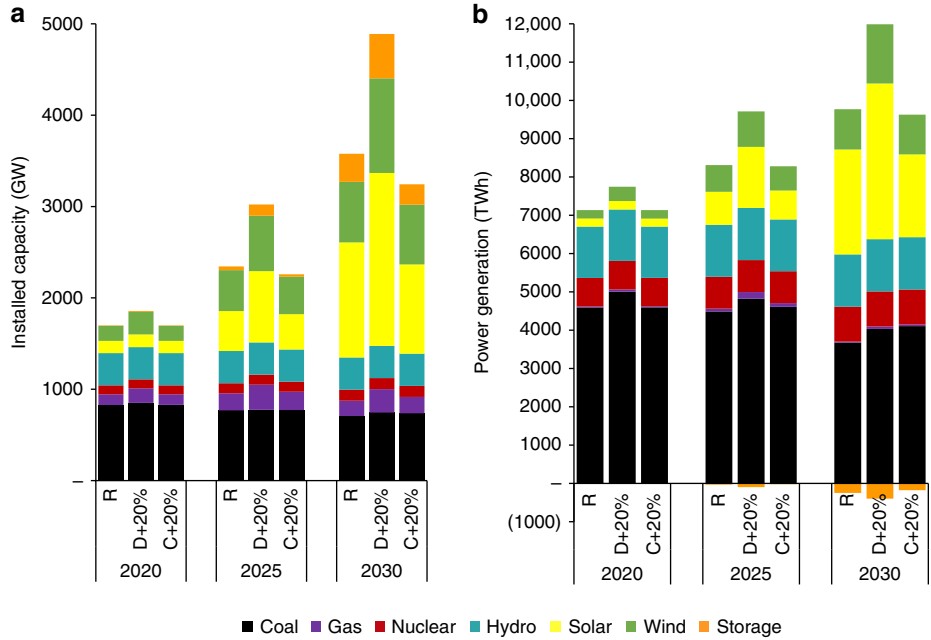

**Fig. 8 Sensitivity analyses of 2030 capacity and generation mixes under the D + 20% and C + 20% scenarios. a** installed capacity mix. **b** power generation mix. D + 20% assumes that demand increases linearly 20% until 2030; C + 20% assumes that the capital costs of solar, wind, and storage are 20% higher than under the R scenario. Source data are provided as a Source Data file.

scenario system requires 307 GW of storage capacity to provide about 250 TWh energy exchange (charge/discharge) and in the C80 scenario about 525 GW of storage capacity to provide about 388 TWh of energy from storage in 2030. Storage is being used

about 2.2 and 2 hours per day to provide the 250 and 388 TWh of storage in the R and C80 scenarios.

Pumped hydro capacity in China in 2015 was about 25 GW, and has been expanding very quickly. It is estimated to have 100 GW,

**Table 1 Model scenarios.**

| | Business as usual (BAU) | Low-cost renewables (R) | Carbon constraints (C50) | Deep carbon constraints (C80) |
|---|---|---|---|---|
| Base year | 2015 | | | |
| Existing policies | Continuation of current policies and no new coal plants after 2020 because of tight regulations on air pollution and institution of carbon mitigation measures[23] | | | |
| Future renewable costs assumptions | Utilizing conventional models for future renewable costs | Rapid decrease in costs for renewables and storage continues: dramatic decreases in wind, solar, and storage costs as projected by Lawrence Berkeley National Laboratory (LBNL) and the National Renewable Energy Laboratory (NREL) | | |
| Carbon constraints | No | No | 50% reduction in power sector $CO_2$ from 2015 level by 2030 | 80% reduction in power sector $CO_2$ from 2015 level by 2030 |

at least 80 GW by 2025, and potentially up to 130 GW by 2030[15]. In this case, to reach 307 GW capacity of storage under the R scenario in 2030, it would require battery storage to reach about 177 GW. With the increase of battery efficiency and performance, the needed storage capacity would be smaller. However, it indeed is very ambitious to deploy such a large scale of storage in a comparatively short time, about 11.8 GW annually during the studying period. Supply chain and life cycle management, economics of storage and policy support are essential to spur the large-scale deployment in order to make such transition happen.

## Discussion

The dramatic decrease in costs for renewable energy enables us to model China's power system and evaluate prospects for accelerating its decarbonization. Our modeling results show that if the costs for solar, wind, and storage follow recent global trends, by 2030 China could derive 62% of needed electricity from non-fossil sources. Total costs under the R scenario are 11% lower than those under the BAU scenario. Under the carbon constraints (C50) scenario, China could eliminate half of its 2015 carbon emissions from the power sector by 2030 with 6% lower cost, while delivering 77% of electricity from non-fossil sources. In the deep carbon constraints (C80) scenario, an 80% emissions reduction from the 2015 level is technically feasible by 2030 but involves about a 21% higher power cost than under the BAU scenario and a $21/tCO_2$ cost of conserved carbon. China has launched a national emissions-trading-system (ETS) with a price range of $3-14.5/t CO_2$, and the carbon price is expected to rise to an average of $16.5/t CO_2$, ranging $4-20/tCO_2$ by 2030[16].

Although modeling results identify possible pathways to accelerate the decarbonization of China's power sector under the four scenarios developed for this study, the speed and scale of expanding the use of renewable energy could be enhanced or impeded by government policies, stakeholder interests, and capital market constraints, among other factors. Positive efforts could include target setting and cost reduction, as exemplified in the renewable portfolio standard in California and elsewhere. Capacity auctions in China and India also create a pricing mechanism to lower the cost of renewables, especially wind and solar. Reforming the power market could create incentives to reduce institutional barriers to trading power across regions and to integrating renewable energy, thereby reducing the curtailment of wind and solar energy observed in the Chinese power sector.

China's power sector is in the midst of expansion and transition. The costs for energy from wind, solar, and storage are affected by many factors such as policy drivers and technological innovation. However, as indicated in the sensitivity analyses, the structural transformation of China's power sector is fairly consistent as long as the cost of renewable technology follows the global trend. This analysis indicates that fast decarbonization of

China's power system is both technically feasible and economically beneficial to China's development, as well as offering the prospect of large emissions mitigation with a global impact.

## Methods

**SWITCH-China model.** To most effectively model the impact of renewables on China's power system, we updated the SWITCH-China capacity expansion model (Supplementary Table 1, Supplementary Note 1, 2). SWITCH, which is a loose acronym for investment in solar, wind, hydro, and conventional technologies, is an optimization model that has the objective function of minimizing the cost of producing and delivering electricity based on projected demand through the construction and retirement of various power generation, storage, and transmission options available currently and at future target dates. The SWITCH-China model provides high resolution in both the temporal and spatial dimensions, to simulate the effect of the dramatically decreasing cost for incorporating renewable energy into the power grid[14]. SWITCH-China runs on a provincial scale and utilizes hourly data to simulate and optimize power system planning based on operational constraints. SWITCH optimizes both the long-term investment and short-term operation of the grid. The model incorporates a combination of current and advanced grid assets. Optimization is subject to reliability, constraints on operations, and resource availability, as well as on current and potential climate policies and environmental regulations[17–21].

SWITCH-China's modeling decisions regarding system expansion are based on optimizing capital costs, operation and maintenance costs, and the variable costs for installed power plant capacities and transmission lines (Supplementary Note 2). Two primary options were available to help us decide which cost projections, from 2015 to 2030, to use in SWITCH-China. LBNL has developed projections of LCOE for utility-scale solar, wind, and storage to 2030. In addition, NREL's latest annual technology baseline (ATB) model projects capacity costs for solar and wind technologies[22]. Although the LCOE is useful in informing investment decisions for many situations, SWITCH-China uses capital, operation and maintenance, and variable costs to develop investment decisions. Thus SWITCH-China assumes that trajectories of capital costs for solar, storage, and wind technologies for the R, C50 and C80 scenarios will resemble NREL's ATB projections to 2030. Trajectories of capital cost for the baseline scenario utilize the original SWITCH-China cost assumptions for advanced technologies to 2030. Except for solar, wind, and storage, all other costs follow the original SWITCH-China cost assumptions (Supplementary Figs. 1 and 2; Supplementary Note 3). The $CO_2$ accounting methods and electricity demand projection are detailed in Supplementary Note 4 and 5.

**Scenarios.** We developed four scenarios in our analysis: business as usual scenario (BAU), low-cost renewables scenario (R), carbon constraints scenario (C50), and deep carbon constraints scenario (C80). Table 1 summarizes the key assumptions of the four scenarios.

**Reporting summary.** Further information on research design is available in the Nature Research Reporting Summary linked to this article.

## Data availability

The source data underlying Figs. 1 to 8 are provided as a Source Data file. All data used for this analysis are available from cited publicly available sources or from the authors upon reasonable request.

## Code availability

Code used in AMPL and Python for this study are available from the authors upon reasonable request.

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

## Acknowledgements

The authors would like to thank Junfeng Hu and David Fridley for their comments. This work was supported by the Energy Foundation China, the Hewlett and MJS Foundation through the U.S. Department of Energy under Contract Number No. DE-AC02-05CH11231.

## Author contributions

G.H. and J.L coordinated the research. G.H., J.L., and A.P. contributed to the study design. G.H. performed the modeling analysis, and led the writing of the paper. G.H., F.S, X.L., and N.A. contributed to the data collection, figure drawing, and policy analysis. All authors provided feedback and contributed to writing the paper.

## Competing interests

The authors declare no competing interests.
