## [Peer Review File · Nature Communications]

Reviewers' comments:

Reviewer #2 (Remarks to the Author):

This manuscript studies the future trends and opportunities for the decarbonization of China's power system through renewable energy and energy storage, with four scenarios considered, including 1) business as usual scenario (BAU), 2) Low-cost renewables scenario (R), 3) Carbon constraints scenario (C50), and 4) Deep carbon constraints scenario (C80). The topic is interesting, however, this paper entails notable technical defects as follows. Therefore, it is recommended not to accept this manuscript in the current form.

Detailed comments:

- 1) This paper lacks a detailed literature review on the renewable potentials, current profile, relevant research methods, which are referred to for this study.
- 2) The section 'Mix of generation capacities and power generation' presents some arbitrary data and information without scientific reasoning and process. The authors should present a scientific procedure with the methodology explained in detail. Based on the procedure, methodology and data used, then the author can explain the results and findings.
- 3) This paper also lacks specific information regarding the potentials and requirements for each type of renewable energy sources, e.g., the installation of wind turbines need to meet certain requirements such as distance from cities and wind speed etc. Similarly, the utilisation of nuclear is also stringently constrained. These constrains are not taken into account in this paper.
- 4) The writing style is far away from scientific writing, e.g, what is the supporting information, such as data or reference, for the following statement?
'The costs of solar photovoltaics (PV), wind, and battery storage have decreased rapidly approximately 65% to 85% since 2010 and are projected to decrease further in the near future '.
- 5) The language is poor with grammar issues and typos. The title should not be a sentence – use highlights and extracted words instead.

Reviewer #3 (Remarks to the Author):

Dear authors, the paper entitled 'Rapid cost decrease of renewable energy and storage offers an opportunity to accelerate the decarbonization of China's power system' is very interesting since it suggests that China can have an aggressive transition to a more sustainable electricity matrix over the next ten years. I believe this study has the potential to be published since it brings novel results and can influence thinking in the field; however, some assumptions made by the authors when running the SWITCH-China model should be clarified. Please find below my main comments.

Line 57. The authors affirm that PV, wind, and battery storage costs have decreased rapidly to approximately 65% to 85% since 2010. Please insert the references and present the assumptions behind those numbers. For example, what are the technological evidence that shows that the projected costs for renewable energy and storage systems would decrease over the next 10 years following the rates assumed in this study?

Line 71. Please clarify the main assumptions considered during the update process. Moreover, this study is based on the SWITCH-China model (reference #10), whose structure is not familiar to everybody. It is not quite clear the difference between this study and the modeling effort within reference # 10. A brief description of the updates and an additional explanation about the basic technical assumptions of the model could also be provided in the supplementary material.

Lines 95-108. The presented results make sense under the statements of the paper. The fact that the cost of renewables would constantly decrease explains the increasing share of renewables in the

energy matrix. However, it lacks a better explanation of how and why the different energy sources increase/decrease their participation in the capacity mix for the four scenarios. In my opinion, the results are too descriptive and lack technical discussion, even if a very brief one. For example, take one type of renewable source (PV, for instance) and then briefly describe which kind of technology evolution justifies an increasing share of this type of renewable source in China's electricity matrix. Does the model consider private and government investment capacities over the next 10 years? As you know, wind and solar power plants are related to very high CAPEX per installed MW. The fact that the future OPEX (mainly fuel cost) is reduced in BAU and R scenarios does not necessarily mean that these savings in brownfield plants would be promptly available to be invested in new greenfield projects. Please clarify how the SWITCH-China model deals with possible limitations in investment capacity until 2030.

Line 101. Typos: 'incaesying', 'capcities', 'storage cpcities'.

Lines 124-130. Indeed, the prevalence of wind and solar as the basis of China's electricity matrix will bring a great challenge when considering the possibility of blackouts/power shortages. The study suggests that batteries and natural gas as a backup plan is the possibility presented in the study, which makes sense. The emissions of such alternatives, however, can be higher when compared to other renewable sources. Were the LCA emissions of such systems considered in the calculations of CO2 mitigation targets of scenarios R50 and R80? Please make it clearer in the manuscript.

Lines 141-151. The discussion in chart 5a and 5b makes total sense, however, it lacks some technical explanation on the assumptions behind the emission and cost reduction curves.

Lines 151-156. Please clarify in the manuscript what the cost of conserved CO2 means, especially when it is negative. What is the reference basis (in terms of energy source) to account for the avoided carbon in China?

Lines 162-171. Please see the question raised in the comments of lines 95-108. Also, it would be interesting to make a parallel calculation on how many power plants should be built every year from 2020 to 2030. Would this number be reasonable given the pace of current investment capacity in the Chinese energy sector?

Lines 175-227. Indeed, there is a very detailed study on the impacts of the scenarios on the transmission system. The magnitude of power lines and distances among regions of the country deserves special attention. Given this, please make clear in the manuscript your assumptions for energy transmission losses and how it is considered in the accountability of carbon emissions.

Figure 9. Something is wrong with the subtitles.

Line 375. Is not the assumption of 1% of the capital costs with O&M costs for renewable energy too low? Please cite one or more references that give support to this assumption.

Figure 11. Please clarify why the capital cost of solar systems is steady (in BAU scenario) whereas storage and wind are related to decreasing costs? For the low-cost scenario, where the assumptions for \$/kW (from 2020-2030) come from?

Response to Reviewers' Comments

We'd like to thank the editor and reviewers for the comments and feedback, we have revised the manuscript and also posted below our point-to-point response to the comments.

Reviewer #2 (Remarks to the Author):

This manuscript studies the future trends and opportunities for the decarbonization of China's power system through renewable energy and energy storage, with four scenarios considered, including 1) business as usual scenario (BAU), 2) Low-cost renewables scenario (R), 3) Carbon constraints scenario (C50), and 4) Deep carbon constraints scenario (C80). The topic is interesting, however, this paper entails notable technical defects as follows. Therefore, it is recommended not to accept this manuscript in the current form.

Detailed comments:

1) This paper lacks a detailed literature review on the renewable potentials, current profile, relevant research methods, which are referred to for this study.

Renewable potentials, current profile, relevant research methods are all very important topics, however, are not the focus of this study. In fact, the author has written two papers to analyze the spatial and temporal availability of solar and wind resources in China^{1,2}:

1. He, Gang, and Daniel M. Kammen. 2014. "Where, When and How Much Wind Is Available? A Provincial-Scale Wind Resource Assessment for China." *Energy Policy* 74: 116–22. <https://doi.org/10.1016/j.enpol.2014.07.003>.
2. He, Gang, and Daniel M. Kammen. 2016. "Where, When and How Much Solar Is Available? A Provincial-Scale Solar Resource Assessment for China." *Renewable Energy* 85: 74–82. <https://doi.org/10.1016/j.renene.2015.06.027>.

The results of these two assessments are used as inputs for provincial solar and wind hourly capacity factors in the current model, in addition, both papers are cited as data source.

2) The section 'Mix of generation capacities and power generation' presents some arbitrary data and information without scientific reasoning and process. The authors should present a scientific procedure with the methodology explained in detail. Based on the procedure, methodology and data used, then the author can explain the results and findings.

The paper was written in the style suggested by *Nature Communications*, that is, we present the results first, and the methods and data in a later section, in which we detailed the description of the SWITCH-China model, and our key assumptions. Please refer to the method and data and the supplementary information sections for more details.

3) This paper also lacks specific information regarding the potentials and requirements for each type of renewable energy sources, e.g., the installation of wind turbines need to meet certain requirements such as distance from cities and wind speed etc. Similarly, the utilisation of nuclear is also stringently constrained. These constrains are not taken into account in this paper.

Similar to our response to comment 1), the resource potential and technical requirements are assessed in our two resource assessment papers, which provide hourly capacity factor at the provincial level for SWITCH-China, the model we use for this paper. With regards to nuclear, we assume China will build as much nuclear capacity as the Chinese government is planning, about 120GW by 2030.³ We take into account all existing, under construction, and planned nuclear capacity that will be online by 2030 into our model. We assume that nuclear plants are run at an average of 85% capacity factor, as reported by China Electricity Council.

4) The writing style is far away from scientific writing, e.g, what is the supporting information, such as data or reference, for the following statement?

‘The costs of solar photovoltaics (PV), wind, and battery storage have decreased rapidly approximately 65% to 85% since 2010 and are projected to decrease further in the near future ‘.

The projection numbers are extracted from IRENA and Bloomberg New Energy Outlook 2019, based on market survey, and consistent with studies from NREL and LBNL. We’ve revised and added the references for this statement.

“The costs of solar photovoltaics (PV), wind, and battery storage have decreased rapidly. The global weighted-average LCOE of utility-scale solar PV, onshore wind, and battery storage has fallen by 77%, 35%, and 85% between 2010 and 2018, respectively.^{4,5}”

Sources:

IRENA. 2019. “Renewable Power Generation Costs in 2018.” Abu Dhabi: International Renewable Energy Agency. https://www.irena.org/-/media/Files/IRENA/Agency/Publication/2019/May/IRENA_Renewable-Power-Generations-Costs-in-2018.pdf.

Logan Goldie-Scot. 2019. Head of Energy Storage. BloombergNEFA. Behind the Scenes Take on Lithium-ion Battery Prices, <https://about.bnef.com/blog/behind-scenes-take-lithium-ion-battery-prices/>

5) The language is poor with grammar issues and typos. The title should not be a sentence – use highlights and extracted words instead.

We have had copy editors work on improving the language, and also proofreading the manuscript to fix any remaining grammar issues and typos. With respect to the title, we decided on a title that captures our key message in the paper. We also follow the Nature Communications’ Guide to authors:

“Title. If possible, this should be 15 words or fewer and should not contain technical terms, abbreviations, punctuation and active verbs.”

Source: <https://www.nature.com/documents/ncomms-submission-guide.pdf>

Reviewer #3 (Remarks to the Author):

Dear authors, the paper entitled ‘Rapid cost decrease of renewable energy and storage offers an opportunity to accelerate the decarbonization of China’s power system’ is very interesting since it suggests that China can have an aggressive transition to a more sustainable electricity matrix over the next ten years. I believe this study has the potential to be published since it brings novel results and can influence thinking in the field; however, some assumptions made by the authors when running the SWITCH-China model should be clarified. Please find below my main comments.

Thank you so much for the detailed comments and suggestions to improve the manuscript. We have incorporated your comments in our revision. Please see our response below.

Line 57. The authors affirm that PV, wind, and battery storage costs have decreased rapidly to approximately 65% to 85% since 2010. Please insert the references and present the assumptions behind those numbers. For example, what are the technological evidence that shows that the projected costs for renewable energy and storage systems would decrease over the next 10 years following the rates assumed in this study?

The projection numbers are extracted from IRENA and Bloomberg New Energy Outlook 2019, based on market survey, and consistent with studies from NREL and LBNL. We’ve added the references for this statement.

“The costs of solar photovoltaics (PV), wind, and battery storage have decreased rapidly. The global weighted-average LCOE of utility-scale solar PV, onshore wind, and battery storage has fallen by 77%, 35%, and 85% between 2010 and 2018, respectively.”

Sources:

IRENA. 2019. “Renewable Power Generation Costs in 2018.” Abu Dhabi: International Renewable Energy Agency. https://www.irena.org/-/media/Files/IRENA/Agency/Publication/2019/May/IRENA_Renewable-Power-Generations-Costs-in-2018.pdf.

Logan Goldie-Scot. 2019. Head of Energy Storage. BloombergNEFA. Behind the Scenes Take on Lithium-ion Battery Prices, <https://about.bnef.com/blog/behind-scenes-take-lithium-ion-battery-prices/>

Project costs for renewables and storage are indeed critical for the analysis. The capital costs assumptions with original data from NREL’s Annual Technology Baseline are posted below^{6,7}:

Year	2015	2016	2017	2018	2019	2020	2021	2022	2023	2024	2025	2026	2027	2028	2029	2030
Solar LBNL Capital Cost (\$/kW)	2380	1874	1627	1234	1130	1189	1065	953	853	726	665	608	557	510	467	427
NREL ATB Onshore wind Capital Cost (\$/kW)	1778	1778	1212	941	893	852	816	793	770	747	724	701	678	654	631	608
NREL ATB Storage (\$/kW) (Source: https://atb.nrel.gov/)	1730	1644	1562	1484	1323	1162	1073	984	896	807	718	672	625	579	532	486

Line 71. Please clarify the main assumptions considered during the update process. Moreover, this study is based on the SWITCH-China model (reference #10), whose structure is not familiar to everybody. It is not quite clear the difference between this study and the modeling effort within reference # 10. A brief description of the updates and an additional explanation about the basic technical assumptions of the model could also be provided in the supplementary material.

We added a summary of the model and key assumptions used. We are also adding a table of comparison of the update in detail in the supplemental information. Key updates include: power plants and transmission lines that were built from 2010-2015 by technology, fuel costs, transmission lines.

Lines 95-108. The presented results make sense under the statements of the paper. The fact that the cost of renewables would constantly decrease explains the increasing share of renewables in the energy matrix. However, it lacks a better explanation of how and why the different energy sources increase/decrease their participation in the capacity mix for the four scenarios. In my opinion, the results are too descriptive and lack technical discussion, even if a very brief one. For example, take one type of renewable source (PV, for instance) and then briefly describe which kind of technology evolution justifies an increasing share of this type of renewable source in China's electricity matrix.

Does the model consider private and government investment capacities over the next 10 years? As you know, wind and solar power plants are related to very high CAPEX per installed MW. The fact that the future OPEX (mainly fuel cost) is reduced in BAU and R scenarios does not necessarily mean that these savings in brownfield plants would be promptly available to be invested in new greenfield projects. Please clarify how the SWITCH-China model deals with possible limitations in investment capacity until 2030.

Considering the resource limits and operation constraints, and other technical and policy constraints, the levelized cost of electricity (LCOE) of different technology at different provinces at different periods is the fundamental driver of where, when and, why certain technologies get built. We added some analysis and technical discussion on the evolution of the LCOE in our model.

Line 101. Typos: 'incaesying', 'capcities', 'storage cpcities'.

We fixed those typos and proofread the whole manuscript to fix any remaining typo and grammar issues.

Lines 124-130. Indeed, the prevalence of wind and solar as the basis of China's electricity matrix will bring a great challenge when considering the possibility of blackouts/power shortages. The study suggests that batteries and natural gas as a backup plan is the possibility presented in the study, which makes sense. The emissions of such alternatives, however, can be higher when compared to other renewable sources. Were the LCA emissions of such systems considered in the calculations of CO2 mitigation targets of scenarios R50 and R80? Please make it clearer in the manuscript.

We added below discussion on storage:

“There is uncertainty around the deployment of large scale storage capacity to integrate renewables. Our results show that in the R scenario, the power system would require 307 GW of storage capacity to provide about 250 TWh of energy exchanges (charge/discharge). In the C80 scenario about 525 GW of storage capacity is needed to provide about 388 TWh of energy from storage in 2030. Storage is being used about 2.2 and 2 hours per day to provide the 250 and 388 TWh of storage in the R and C80 scenarios.

Pumped hydro capacity in China in 2015 was about 25 GW, and has been expanding very quickly. It is estimated to have 100 GW, at least 80 GW by 2025, and potentially up to 130 GW by 2030.⁸ In this case, assuming these pumped hydro installed capacities, to reach 307 GW capacity of storage under the R scenario in 2030, it would require a 177 GW of battery storage. With the increase of battery efficiency and performance, the needed storage capacity is expected to be smaller. Deploying storage capacity in such a large scale, in a comparatively short time, would amount to an ambitious annual 11.8 GW capacity added during the period studied. Supply chain and life cycle management, economics of storage and policy support are essential to spur the large-scale deployment in order to make such transition happen.”

The paper uses the fuel emission factor as guided by China’s National Development and Reform Commission⁹ for emission calculation as posted below:

Coal: 25.41 kgC/GJ (93.17 kgCO₂/GJ)

Natural Gas: 15.32 kgC/GJ (56.22 kgCO₂/GJ)

Oil: 21.10 kgC/GJ (77.37 kgCO₂/GJ)

We did not consider life-cycle emissions of such systems rather focusing on the emissions of fuel combustions for two reasons: First, the LCA emissions of solar and wind technologies are comparatively small, mean value reported at 34.1gCO₂-eq/kWh and 49.9gCO₂-eq/kWh for wind and solar, respectively, according to a meta review.¹⁰ Second, the Intended Nationally Determined Contributions (INDCs), used as baselines for this study, have not yet included LCA emissions calculations.

Source: NDRC, China Provincial GHG Emission Inventory Guideline. 2011.

<http://www.cbcsd.org.cn/sjk/nengyuan/standard/home/20140113/download/shengjiwenshiqiti.pdf>

Nugent, Daniel, and Benjamin K. Sovacool. 2014. “Assessing the Lifecycle Greenhouse Gas Emissions from Solar PV and Wind Energy: A Critical Meta-Survey.” *Energy Policy* 65 (February): 229–44. <https://doi.org/10.1016/j.enpol.2013.10.048>.

Lines 141-151. The discussion in chart 5a and 5b makes total sense, however, it lacks some technical explanation on the assumptions behind the emission and cost reduction curves.

The emissions from all fuel combustion to generate electricity to meet future demand are summarized, and the costs are the total costs to supply electricity demand under different scenarios. We added the assumptions in the supplementary information.

Lines 151-156. Please clarify in the manuscript what the cost of conserved CO₂ means, especially when it is negative. What is the reference basis (in terms of energy source) to account for the avoided carbon in China?

Cost of conserved CO₂ in this paper means the costs per ton of mitigated CO₂ compared to the reference scenario. This indicator shows the extra cost to achieve deeper decarbonization compared to business as usual. A negative cost means it is actually cheaper to achieve those carbon mitigations due to the decline of renewable and storage costs.

Lines 162-171. Please see the question raised in the comments of lines 95-108. Also, it would be interesting to make a parallel calculation on how many power plants should be built every year from 2020 to 2030. Would this number be reasonable given the pace of current investment capacity in the Chinese energy sector?

We added some analysis to incorporate your suggestions. However, we try to make objective comparison, rather than draw from what may seem arbitrary reasoning to the readers.

Lines 151-155: “The cost of conserved CO₂ would be -\$36/tCO₂, -\$9/tCO₂, and \$21/tCO₂ under the R scenario, C50 scenario, and C80 scenario, respectively. China has already initiated a national cap-and-trade program limiting the carbon emissions from the power sector with a carbon price ranging from 20 RMB/tCO₂ (\$3/tCO₂) to 100 RMB/tCO₂ (\$14.5/tCO₂).”

Lines 175-227. Indeed, there is a very detailed study on the impacts of the scenarios on the transmission system. The magnitude of power lines and distances among regions of the country deserves special attention. Given this, please make clear in the manuscript your assumptions for energy transmission losses and how it is considered in the accountability of carbon emissions.

We added the transmission assumptions into the supplemental information. The average transmission loss is assumed to be 6%, according to the China Electricity Quick Statistic 2019 published by China Electricity Council.¹¹

Figure 9. Something is wrong with the subtitles.

We checked the subtitles and make sure those are properly displayed.

Line 375. Is not the assumption of 1% of the capital costs with O&M costs for renewable energy too low? Please cite one or more references that give support to this assumption.

Figure 11. Please clarify why the capital cost of solar systems is steady (in BAU scenario) whereas storage and wind are related to decreasing costs? For the low-cost scenario, where the assumptions for \$/kW (from 2020-2030) come from?

We fixed those typos and proofread the whole manuscript to fix any remaining typo and grammar issues. The operation and maintenance costs are assumed to be 1% of the capital costs, which are at equivalent level of those by NREL Annual Technology Baseline.⁶ The cost of solar PV in the BAU scenario are project to 2020 provided by the SunShot Initiative and then stays at

the 2020 level.¹² Project costs for renewables and storage are indeed critical for the analysis. The capital costs assumptions with original data are posted below:

Year	2015	2016	2017	2018	2019	2020	2021	2022	2023	2024	2025	2026	2027	2028	2029	2030
Solar LBNL Capital Cost (\$/kW)	2380	1874	1627	1234	1130	1189	1065	953	853	726	665	608	557	510	467	427
NREL ATB Onshore wind Capital Cost (\$/kW)	1778	1778	1212	941	893	852	816	793	770	747	724	701	678	654	631	608
NREL ATB Storage (\$/kW) (Source: https://atb.nrel.gov/)	1730	1644	1562	1484	1323	1162	1073	984	896	807	718	672	625	579	532	486

References:

1. He, G. & Kammen, D. M. Where, when and how much wind is available? A provincial-scale wind resource assessment for China. *Energy Policy* **74**, 116–122 (2014).
2. He, G. & Kammen, D. M. Where, when and how much solar is available? A provincial-scale solar resource assessment for China. *Renewable Energy* **85**, 74–82 (2016).
3. CNDC & SGERI. *National Nuclear Development Plan Research*.
<http://www.ccchina.org.cn/Detail.aspx?newsId=28029&TId=60> (2019).
4. IRENA. *Renewable power generation costs in 2018*. 88 https://www.irena.org/-/media/Files/IRENA/Agency/Publication/2019/May/IRENA_Renewable-Power-Generations-Costs-in-2018.pdf (2019).
5. BloombergNEF. *New Energy Outlook 2019*. <https://about.bnef.com/new-energy-outlook/#toc-download> (2019).
6. NREL. *2018 Annual Technology Baseline*. <https://data.nrel.gov/files/89/2018-ATB-data-interim-geo.xlsx> (2018).
7. NREL. *2019 Annual Technology Baseline*. <https://atb.nrel.gov/electricity/2019/files/2019-ATB-data.xlsx> (2019).
8. Jia, K. Zheng Sheng'an: 2030 Wind and Solar Capacity Will Reach 1200GW. *China Energy News* (2019).

9. NDRC. *China Provincial GHG Emission Inventory Guideline*.
<http://www.cbcsd.org.cn/sjk/nengyuan/standard/home/20140113/download/shengjiwenshiqiti.pdf> (2011).
10. Nugent, D. & Sovacool, B. K. Assessing the lifecycle greenhouse gas emissions from solar PV and wind energy: A critical meta-survey. *Energy Policy* **65**, 229–244 (2014).
11. CEC. *China Electricity Industry Quick Statistic 2019*.
<http://www.cec.org.cn/d/file/guihuayutongji/tongjixinxi/niandushuju/2020-01-21/da4b94b0ea26eb47bb0304bc44970870.pdf> (2020).
12. DOE. *SunShot Vision Study*. <http://energy.gov/sites/prod/files/2014/01/f7/47927.pdf> (2012).

Reviewers' comments:

Reviewer #2 (Remarks to the Author):

The authors have addressed the raised comments, and the supplementary document is useful to help the readers to understand the model and the difference from the published work. It is recommended to accept this manuscript.

Reviewer #3 (Remarks to the Authors):

Dear authors, I believe most of the points raised by me were properly addressed. The inclusion of the supplementary material is very important to give a more solid explanation to the readers of your paper, especially for those who doesn't know SWITCH-China model. I still believe this paper has potential for publication; however, there were some minor aspects that were not fully explained in my opinion (GHG accountability and technological explanation for cost reduction pointed out in your references). Please find below my reply for all of your answers point-by-point.

Thank you so much for the detailed comments and suggestions to improve the manuscript. We have incorporated your comments in our revision. Please see our response below.

Line 57. The authors affirm that PV, wind, and battery storage costs have decreased rapidly to approximately 65% to 85% since 2010. Please insert the references and present the assumptions behind those numbers. For example, what are the technological evidence that shows that the projected costs for renewable energy and storage systems would decrease over the next 10 years following the rates assumed in this study?

The projection numbers are extracted from IRENA and Bloomberg New Energy Outlook 2019, based on market survey, and consistent with studies from NREL and LBNL. We've added the references for this statement.

“The costs of solar photovoltaics (PV), wind, and battery storage have decreased rapidly. The global weighted-average LCOE of utility-scale solar PV, onshore wind, and battery storage has fallen by 77%, 35%, and 85% between 2010 and 2018, respectively.”

Sources:

IRENA. 2019. “Renewable Power Generation Costs in 2018.” Abu Dhabi: International Renewable Energy Agency. https://www.irena.org/-/media/Files/IRENA/Agency/Publication/2019/May/IRENA_Renewable-Power-Generations-Costs-in-2018.pdf.

Logan Goldie-Scot. 2019. Head of Energy Storage. BloombergNEFA. Behind the Scenes Take on Lithium-ion Battery Prices, <https://about.bnef.com/blog/behind-scenes-take-lithium-ion-battery-prices/>

Project costs for renewables and storage are indeed critical for the analysis. The capital costs assumptions with original data from NREL's Annual Technology Baseline are posted below^{6,7}:

Year	2015	2016	2017	2018	2019	2020	2021	2022	2023	2024	2025	2026	2027	2028	2029	2030
Solar LBNL Capital Cost (\$/kW)	2380	1874	1627	1234	1130	1189	1065	953	853	726	665	608	557	510	467	427
NREL ATB Onshore wind Capital Cost (\$/kW)	1778	1778	1212	941	893	852	816	793	770	747	724	701	678	654	631	608
NREL ATB Storage (\$/kW) (Source: https://atb.nrel.gov/)	1730	1644	1562	1484	1323	1162	1073	984	896	807	718	672	625	579	532	486

Ok. I believe the question was answered properly. Although the study from BloombergNEFA is market-based study instead of a scientific one, there might be a deep survey effort behind those assumptions for battery storage cost learning curve. In my point of view, the information is good enough to build a future scenario.

Line 71. Please clarify the main assumptions considered during the update process. Moreover, this study is based on the SWITCH-China model (reference #10), whose structure is not familiar to everybody. It is not quite clear the difference between this study and the modeling effort within reference # 10. A brief description of the updates and an additional explanation about the basic technical assumptions of the model could also be provided in the supplementary material.

We added a summary of the model and key assumptions used. We are also adding a table of comparison of the update in detail in the supplemental information. Key updates include: power plants and transmission lines that were built from 2010-2015 by technology, fuel costs, transmission lines.

Ok, good. I believe those explanations in the supplementary information will help readers to understand the main structure of the model.

Lines 95-108. The presented results make sense under the statements of the paper. The fact that the cost of renewables would constantly decrease explains the increasing share of renewables in the energy matrix. However, it lacks a better explanation of how and why the different energy sources increase/decrease their participation in the capacity mix for the four scenarios. In my opinion, the results are too descriptive and lack technical discussion, even if a very brief one. For example, take one type of renewable source (PV, for instance) and then briefly describe which kind of technology evolution justifies an increasing share of this type of renewable source in China's electricity matrix.

Does the model consider private and government investment capacities over the next 10 years? As you know, wind and solar power plants are related to very high CAPEX per installed MW. The fact that the future OPEX (mainly fuel cost) is reduced in BAU and R scenarios does not necessarily mean that these savings in brownfield plants would be promptly available to be invested in new greenfield projects. Please clarify how the SWITCH-China model deals with possible limitations in investment capacity until 2030.

Considering the resource limits and operation constraints, and other technical and policy constraints, the levelized cost of electricity (LCOE) of different technology at different provinces at different periods is the fundamental driver of where, when and, why certain technologies get built. We added some analysis and technical discussion on the evolution of the LCOE in our model.

Ok. When reading the explanation about the SWITCH-China model it is easier to comprehend now that these constraints are taken into account as you have an optimization/minimization algorithm in your simulations.

Line 101. Typos: 'inceeding', 'capcities', 'storage cpcities'.

We fixed those typos and proofread the whole manuscript to fix any remaining typo and grammar issues.

Ok.

Lines 124-130. Indeed, the prevalence of wind and solar as the basis of China's electricity matrix will bring a great challenge when considering the possibility of blackouts/power shortages. The study suggests that batteries and natural gas as a backup plan is the possibility presented in the study, which makes sense. The emissions of such alternatives, however, can be higher when compared to other renewable sources. Were the LCA emissions of such systems considered in the calculations of CO₂ mitigation targets of scenarios R50 and R80? Please make it clearer in the manuscript.

We added below discussion on storage:

“There is uncertainty around the deployment of large scale storage capacity to integrate renewables. Our results show that in the R scenario, the power system would require 307 GW of storage capacity to provide about 250 TWh of energy exchanges (charge/discharge). In the C80 scenario about 525 GW of storage capacity is needed to provide about 388 TWh of energy from storage in 2030. Storage is being used about 2.2 and 2 hours per day to provide the 250 and 388 TWh of storage in the R and C80 scenarios.

Pumped hydro capacity in China in 2015 was about 25 GW, and has been expanding very quickly. It is estimated to have 100 GW, at least 80 GW by 2025, and potentially up to 130 GW by 2030.⁸ In this case, assuming these pumped hydro installed capacities, to reach 307 GW capacity of storage under the R scenario in 2030, it would require a 177 GW of battery storage. With the increase of battery efficiency and performance, the needed storage capacity is expected to be smaller. Deploying storage capacity in such a large scale, in a comparatively short time, would amount to an ambitious annual 11.8 GW capacity added during the period studied. Supply chain and life cycle management, economics of storage and policy support are essential to spur the large-scale deployment in order to make such transition happen.”

The paper uses the fuel emission factor as guided by China's National Development and Reform Commission⁹ for emission calculation as posted below:

Coal: 25.41 kgC/GJ (93.17 kgCO₂/GJ)

Natural Gas: 15.32 kgC/GJ (56.22 kgCO₂/GJ)

Oil: 21.10 kgC/GJ (77.37 kgCO₂/GJ)

We did not consider life-cycle emissions of such systems rather focusing on the emissions of fuel combustions for two reasons: First, the LCA emissions of solar and wind technologies are comparatively small, mean value reported at 34.1gCO₂-eq/kWh and 49.9gCO₂-eq/kWh for wind and solar, respectively, according to a meta review.¹⁰ Second, the Intended Nationally Determined Contributions (INDCs), used as baselines for this study, have not yet included LCA emissions calculations.

Source: NDRC, China Provincial GHG Emission Inventory Guideline. 2011.

<http://www.cbcsd.org.cn/sjk/nengyuan/standard/home/20140113/download/shengjiwenshiqiti.pdf>

Nugent, Daniel, and Benjamin K. Sovacool. 2014. "Assessing the Lifecycle Greenhouse Gas Emissions from Solar PV and Wind Energy: A Critical Meta-Survey." *Energy Policy* 65 (February): 229–44. <https://doi.org/10.1016/j.enpol.2013.10.048>.

Indeed, LCA emissions of renewable sources are lower than fossil. On the other hand, the fuel combustion factor is an incomplete metric since it doesn't consider the emissions from the entire carbon footprint. Another observation is related to the emissions of fossil sources. For example, life-cycle emissions of electricity generation from natural gas, oil and coal, at an average power plant, range from 200-300 gCO₂-eq per MJ (these values are much higher than the values from your reference 56-93 gCO₂/MJ). The reference #9 is not in English (it is in Chinese, instead) and it was difficult to understand the calculations and your assumptions in this case. Even not considering the LCA emissions, please clarify the underlying assumptions as an additional item in the supplementary material; that will help readers to understand how the calculations of % in carbon reduction were obtained (it refers to Figure 5, manuscript).

Lines 141-151. The discussion in chart 5a and 5b makes total sense, however, it lacks some technical explanation on the assumptions behind the emission and cost reduction curves.

The emissions from all fuel combustion to generate electricity to meet future demand are summarized, and the costs are the total costs to supply electricity demand under different scenarios. We added the assumptions in the supplementary information.

The inclusion of the supplementary is very important and should be maintained. However, there is no technical explanation for the cost reduction curves. For example, see Figure 4. The only explanation you provide is that the cost reduction will follow historical trend. I believe this is not enough to extrapolate such costs to the future. For example, some additional information from NREL that justifies the capital cost reduction using technical explanation about PV, wind, and storage technologies. And what, technically speaking, justifies the huge cost reduction from BAU to Low Cost scenarios (fig. 4 for example)?

Lines 151-156. Please clarify in the manuscript what the cost of conserved CO₂ means, especially when it is negative. What is the reference basis (in terms of energy source) to account for the avoided carbon in China?

Cost of conserved CO₂ in this paper means the costs per ton of mitigated CO₂ compared to the reference scenario. This indicator shows the extra cost to achieve deeper decarbonization compared to business as usual. A negative cost means it is actually cheaper to achieve those carbon mitigations due to the decline of renewable and storage costs.

Ok.

Lines 162-171. Please see the question raised in the comments of lines 95-108. Also, it would be interesting to make a parallel calculation on how many power plants should be built every year from 2020 to 2030. Would this number be reasonable given the pace of current investment capacity in the Chinese energy sector?

We added some analysis to incorporate your suggestions. However, we try to make objective comparison, rather than draw from what may seem arbitrary reasoning to the readers. Lines 151-155: “The cost of conserved CO₂ would be -\$36/tCO₂, -\$9/tCO₂, and \$21/tCO₂ under the R scenario, C50 scenario, and C80 scenario, respectively. China has already initiated a national cap-and-trade program limiting the carbon emissions from the power sector with a carbon price ranging from 20 RMB/tCO₂ (\$3/tCO₂) to 100 RMB/tCO₂ (\$14.5/tCO₂).”

Ok.

Lines 175-227. Indeed, there is a very detailed study on the impacts of the scenarios on the transmission system. The magnitude of power lines and distances among regions of the country deserves special attention. Given this, please make clear in the manuscript your assumptions for energy transmission losses and how it is considered in the accountability of carbon emissions.

We added the transmission assumptions into the supplemental information. The average transmission loss is assumed to be 6%, according to the China Electricity Quick Statistic 2019 published by China Electricity Council.¹¹

Ok.

Figure 9. Something is wrong with the subtitles.

We checked the subtitles and make sure those are properly displayed.

Ok.

Line 375. Is not the assumption of 1% of the capital costs with O&M costs for renewable energy too low? Please cite one or more references that give support to this assumption.

Figure 11. Please clarify why the capital cost of solar systems is steady (in BAU scenario) whereas storage and wind are related to decreasing costs? For the low-cost scenario, where the assumptions for \$/kW (from 2020-2030) come from?

We fixed those typos and proofread the whole manuscript to fix any remaining typo and grammar issues. The operation and maintenance costs are assumed to be 1% of the capital costs, which are at equivalent level of those by NREL Annual Technology Baseline.⁶ The cost of solar PV in the BAU scenario are project to 2020 provided by the SunShot Initiative and then stays at the 2020 level.¹² Project costs for renewables and storage are indeed critical for the analysis. The capital costs assumptions with original data are posted below:

Year	2015	2016	2017	2018	2019	2020	2021	2022	2023	2024	2025	2026	2027	2028	2029	2030
Solar LBNL Capital Cost (\$/kW)	2380	1874	1627	1234	1130	1189	1065	953	853	726	665	608	557	510	467	427
NREL ATB Onshore wind Capital Cost (\$/kW)	1778	1778	1212	941	893	852	816	793	770	747	724	701	678	654	631	608
NREL ATB Storage (\$/kW) (Source: https://atb.nrel.gov/)	1730	1644	1562	1484	1323	1162	1073	984	896	807	718	672	625	579	532	486

Ok.

References:

1. He, G. & Kammen, D. M. Where, when and how much wind is available? A provincial-scale wind resource assessment for China. *Energy Policy* **74**, 116–122 (2014).
2. He, G. & Kammen, D. M. Where, when and how much solar is available? A provincial-scale solar resource assessment for China. *Renewable Energy* **85**, 74–82 (2016).
3. CNDC & SGERI. *National Nuclear Development Plan Research*.
<http://www.ccchina.org.cn/Detail.aspx?newsId=28029&Tid=60> (2019).
4. IRENA. *Renewable power generation costs in 2018*. 88 https://www.irena.org/-/media/Files/IRENA/Agency/Publication/2019/May/IRENA_Renewable-Power-Generations-Costs-in-2018.pdf (2019).
5. BloombergNEF. *New Energy Outlook 2019*. <https://about.bnef.com/new-energy-outlook/#toc-download> (2019).
6. NREL. *2018 Annual Technology Baseline*. <https://data.nrel.gov/files/89/2018-ATB-data-interim-geo.xlsx> (2018).
7. NREL. *2019 Annual Technology Baseline*. <https://atb.nrel.gov/electricity/2019/files/2019-ATB-data.xlsx> (2019).
8. Jia, K. Zheng Sheng'an: 2030 Wind and Solar Capacity Will Reach 1200GW. *China Energy News* (2019).
9. NDRC. *China Provincial GHG Emission Inventory Guideline*.
<http://www.cbcsd.org.cn/sjk/nengyuan/standard/home/20140113/download/shengjiwenshiqiti.pdf> (2011). **this reference is not in English.**
10. Nugent, D. & Sovacool, B. K. Assessing the lifecycle greenhouse gas emissions from solar PV and wind energy: A critical meta-survey. *Energy Policy* **65**, 229–244 (2014).

11. CEC. *China Electricity Industry Quick Statistic 2019*.
<http://www.cec.org.cn/d/file/guihuayutongji/tongjixinxi/niandushuju/2020-01-21/da4b94b0ea26eb47bb0304bc44970870.pdf> (2020). **this reference is not in English.**
12. DOE. *SunShot Vision Study*. <http://energy.gov/sites/prod/files/2014/01/f7/47927.pdf>
(2012).

Response to reviewer comments

Reviewer #3 (Remarks to the Authors):

Dear authors, I believe most of the points raised by me were properly addressed. The inclusion of the supplementary material is very important to give a more solid explanation to the readers of your paper, especially for those who doesn't know SWITCH-China model. I still believe this paper has potential for publication; however, there were some minor aspects that were not fully explained in my opinion (GHG accountability and technological explanation for cost reduction pointed out in your references). Please find below my reply for all of your answers point-by-point.

Dear reviewer, we appreciated your insightful comments which helped us to improve the manuscript. Please see our point-to-point response to your remaining comments.

Thank you so much for the detailed comments and suggestions to improve the manuscript. We have incorporated your comments in our revision. Please see our response below.

Line 57. The authors affirm that PV, wind, and battery storage costs have decreased rapidly to approximately 65% to 85% since 2010. Please insert the references and present the assumptions behind those numbers. For example, what are the technological evidence that shows that the projected costs for renewable energy and storage systems would decrease over the next 10 years following the rates assumed in this study?

The historical numbers are extracted from IRENA and Bloomberg New Energy Outlook 2019, based on market survey, and consistent with studies from NREL and LBNL. We've added the references for this statement.

“The costs of solar photovoltaics (PV), wind, and battery storage have decreased rapidly. The global weighted-average LCOE of utility-scale solar PV, onshore wind, and battery storage has fallen by 77%, 35%, and 85% between 2010 and 2018, respectively.”

Sources:

IRENA. 2019. “Renewable Power Generation Costs in 2018.” Abu Dhabi: International Renewable Energy Agency. https://www.irena.org/-/media/Files/IRENA/Agency/Publication/2019/May/IRENA_Renewable-Power-Generations-Costs-in-2018.pdf.

Logan Goldie-Scot. 2019. Head of Energy Storage. BloombergNEFA. Behind the Scenes Take on Lithium-ion Battery Prices, <https://about.bnef.com/blog/behind-scenes-take-lithium-ion-battery-prices/>

Future projected costs for renewables and storage are indeed critical for the analysis. The capital costs assumptions with original data from NREL's Annual Technology Baseline are posted below^{1,2}:

Year	2015	2016	2017	2018	2019	2020	2021	2022	2023	2024	2025	2026	2027	2028	2029	2030
Solar LBNL Capital Cost (\$/kW)	2380	1874	1627	1234	1130	1189	1065	953	853	726	665	608	557	510	467	427
NREL ATB Onshore wind Capital Cost (\$/kW)	1778	1778	1212	941	893	852	816	793	770	747	724	701	678	654	631	608
NREL ATB Storage (\$/kW) (Source: https://atb.nrel.gov/)	1730	1644	1562	1484	1323	1162	1073	984	896	807	718	672	625	579	532	486

Ok. I believe the question was answered properly. Although the study from BloombergNEFA is market-based study instead of a scientific one, there might be a deep survey effort behind those assumptions for battery storage cost learning curve. In my point of view, the information is good enough to build a future scenario.

The solar and on shore wind costs reduction are based on the IRENA’s most recent cost report using their global database.

Pg 13 of the report: “In 2018, around 60 GW of new utility-scale solar PV was commissioned (with another 34 GW of residential and commercial rooftop solar PV added). The utility-scale solar PV projects commissioned in 2018 had a global weighted-average LCOE of USD 0.085/kWh, which was around 13% lower than the equivalent figure for 2017. The global weighted-average **LCOE of utility-scale solar PV has fallen by 77% between 2010 and 2018.**”

Pg 19 of the report: “The global weighted-average LCOE of onshore wind projects commissioned in 2018, at USD 0.056/kWh, was 13% lower than in 2017 and **35% lower than in 2010**, when it was USD 0.085/kWh. Costs of electricity from onshore wind are now at the lower end of the fossil fuel cost range.”

BloombergNEFA report offers some recent market insights which are consistent (in similar range) with the IRENA study.

IRENA report does not report the costs of storage, we found BloombergNEFA offer some observation. “The annual price survey has become an important benchmark in the industry and the fall in prices has been nothing short of remarkable: **the volume weighted average battery pack fell 85% from 2010-18**, reaching an average of \$176/kWh.” While BloombergNEFA is market-based survey rather than scientific study, it is widely quoted in the industry.

In addition, those are all historical costs, and were only used to indicate the fast costs drop and are not the price projections used in the model. The projections are based on LBNL and NREL’s Annual Technology Baseline reports. Costs projection itself is very important and in fact is another good research topic that need more efforts.

Lines 124-130. Indeed, the prevalence of wind and solar as the basis of China’s electricity matrix will bring a great challenge when considering the possibility of blackouts/power shortages. The study suggests that batteries and natural gas as a backup plan is the possibility presented in the study, which makes sense. The emissions of such alternatives, however, can be higher when compared to other renewable sources. Were the LCA emissions of such systems considered in the calculations of CO2 mitigation targets of scenarios R50 and R80? Please make it clearer in the manuscript.

We added below discussion on storage:

“There is uncertainty around the deployment of large scale storage capacity to integrate renewables. Our results show that in the R scenario, the power system would require 307 GW of storage capacity to provide about 250 TWh of energy exchanges (charge/discharge). In the C80 scenario about 525 GW of storage capacity is needed to provide about 388 TWh of energy from storage in 2030. Storage is being used about 2.2 and 2 hours per day to provide the 250 and 388 TWh of storage in the R and C80 scenarios.

Pumped hydro capacity in China in 2015 was about 25 GW, and has been expanding very quickly. It is estimated to have 100 GW, at least 80 GW by 2025, and potentially up to 130 GW by 2030.³ In this case, assuming these pumped hydro installed capacities, to reach 307 GW capacity of storage under the R scenario in 2030, it would require a 177 GW of battery storage. With the increase of battery efficiency and performance, the needed storage capacity is expected to be smaller. Deploying storage capacity in such a large scale, in a comparatively short time, would amount to an ambitious annual 11.8 GW capacity added during the period studied. Supply chain and life cycle management, economics of storage and policy support are essential to spur the large-scale deployment in order to make such transition happen.”

The paper uses the fuel emission factor as guided by China’s National Development and Reform Commission⁴ for emission calculation as posted below:

Coal: 25.41 kgC/GJ (93.17 kgCO₂/GJ)

Natural Gas: 15.32 kgC/GJ (56.22 kgCO₂/GJ)

Oil: 21.10 kgC/GJ (77.37 kgCO₂/GJ)

We did not consider life-cycle emissions of such systems rather focusing on the emissions of fuel combustions for two reasons: First, the LCA emissions of solar and wind technologies are comparatively small, mean value reported at 34.1gCO₂-eq/kWh and 49.9gCO₂-eq/kWh for wind and solar, respectively, according to a meta review.⁵ Second, the Intended Nationally Determined Contributions (INDCs), used as baselines for this study, have not yet included LCA emissions calculations.

Source: NDRC, China Provincial GHG Emission Inventory Guideline. 2011.

<http://www.cbcsd.org.cn/sjk/nengyuan/standard/home/20140113/download/shengjiwenshiqiti.pdf>

Nugent, Daniel, and Benjamin K. Sovacool. 2014. “Assessing the Lifecycle Greenhouse Gas Emissions from Solar PV and Wind Energy: A Critical Meta-Survey.” *Energy Policy* 65 (February): 229–44. <https://doi.org/10.1016/j.enpol.2013.10.048>.

Indeed, LCA emissions of renewable sources are lower than fossil. On the other hand, the fuel combustion factor is an incomplete metric since it doesn’t consider the emissions from the entire carbon footprint. Another observation is related to the emissions of fossil sources. For example, life-cycle emissions of electricity generation from natural gas, oil and coal, at an average power plant, range from 200-300 gCO₂-eq per MJ (these values are much higher than the values from your reference 56-93 gCO₂/MJ). The reference #9 is not in English (it is in Chinese, instead) and it was difficult to understand the calculations and your assumptions in this case. Even not considering the LCA emissions, please clarify the underlying assumptions as an additional item in the supplementary material; that will help readers to understand how the calculations of % in carbon reduction were obtained (it refers to Figure 5, manuscript).

Thank you for the suggestion, we added a session in the supplementary information to clarify our assumptions and discuss LCA emissions and their implications to the results. We also updated all the Chinese reference with a blanket (in Chinese) to indicate the source. For this specific source, this guideline provided by China’s National Development and Reform Commission (NDRC) for provincial and local governments to create their emission inventory, which is already adopted in the practice and therefore are reasonable numbers to use. Page 15 of the guideline shows the emission factor by fuel type at different industrial processes due to difference on carbohydrate oxidation rate. We choose the numbers of power and heat industry by fuel type. The calculation is based on the carbon accounting and carbon constraint in SWITCH-China model and also copied below.

Carbon Accounting/Cap Constraint

This constraint requires that, for every period, the total carbon dioxide emissions from generation and spinning reserve provision cannot exceed a pre-specified emission cap. Emissions are incurred for power generation, provision of spinning reserves, cycling of plants below full load, and generator start-up.

$ \begin{aligned} & CARBON_CAP_i \\ & \sum_{p,t \in T_i} O_{p,t} \times hr_p \times CO_{2f_p} + \\ & \sum_{p \in DPUIP, t \in T_i} SP_{p,t} \times sp_penalty_p \times CO_{2f_p} + \\ & \sum_{p \in FBPUIP, t \in T_i} DC_{p,t} \times dc_penalty_p \times \\ & CO_{2f_p} + \\ & \sum_{p \in DPUIP, t \in T_i} ST_{p,t} \times startup_fuel_p \times \\ & CO_{2f_p} \leq carbon_cap_i \end{aligned} $	In every period i, the total carbon emissions cannot exceed a pre-specified carbon cap $carbon_cap_i$ for that period. Total carbon emissions are incurred from generation (calculated as the plant output $O_{p,t}$ times the plant heat rate hr_p times the carbon dioxide fuel content for that plant); plus the carbon emissions from spinning reserve from dispatchable and intermediate plants (calculated as the amount of spinning reserves provided $SP_{p,t}$ times the plant per unit heat rate penalty for providing spinning reserve $sp_penalty_p$ times the CO₂ fuel content for that plant); plus the carbon emissions from deep-cycling flexible baseload and intermediate plants below full load (calculated as the amount below full load $DC_{p,t}$ times the heat rate penalty for cycling below full load $dc_penalty_p$ times the CO₂ fuel content); plus the emissions from starting up intermediate and dispatchable plants (calculated as the capacity started up since the previous hour $ST_{p,t}$ times the startup fuel required $startup_fuel_p$ times the CO₂ fuel content).
--	--

We added below information in the supplementary information:

“3. CO₂ accounting

The paper uses the fuel emission factors as provided in the *China Provincial GHG Emission Inventory Guideline (in Chinese)* released by China’s National Development and Reform Commission (NDRC) for emission calculation as posted below:

Coal: 25.41 kgC/GJ (93.17 kgCO₂/GJ)

Natural Gas: 15.32 kgC/GJ (56.22 kgCO₂/GJ)

Oil: 21.10 kgC/GJ (77.37 kgCO₂/GJ)

The total emissions are calculated with sum of plant level emissions from generation and spinning reserve provision and cannot exceed a pre-specified emission cap if a carbon constraint is introduced.⁶

We are aware of the importance of the life-cycle assessment (LCA) emissions of different technologies. The mean value of LCA emissions of solar and wind technologies are reported at 34.1gCO₂-eq/kWh and 49.9gCO₂-eq/kWh for wind and solar, respectively, according to a meta review.⁵ In this study, we focus on the direct emissions so to make it more comparable to China’s existing carbon mitigation goals. China’s Intended Nationally Determined Contributions (INDCs) and existing carbon mitigation goals have not incorporated life-cycle carbon emissions.⁷ Future studies are needed to address the question on how LCA emissions would impact power capacity expansion.”

Lines 141-151. The discussion in chart 5a and 5b makes total sense, however, it lacks some technical explanation on the assumptions behind the emission and cost reduction curves.

The emissions from all fuel combustion to generate electricity to meet future demand are summarized, and the costs are the total costs to supply electricity demand under different scenarios. We added the assumptions in the supplementary information.

The inclusion of the supplementary is very important and should be maintained. However, there is no technical explanation for the cost reduction curves. For example, see Figure 4. The only explanation you provide is that the cost reduction will follow historical trend. I believe this is not enough to extrapolate such costs to the future. For example, some additional information from NREL that justifies the capital cost reduction using technical explanation about PV, wind, and storage technologies. And what, technically speaking, justifies the huge cost reduction from BAU to Low Cost scenarios (fig. 4 for example)?

Thank you for the suggestion. The projections are based on LBNL research and NREL's Annual Technology Baseline reports. Costs projection itself is very important and in fact is another good research topic that need further efforts. We added below clarification and justification underline the NREL and LBNL's projection in the supplementary information to support our choice of cost projections.

“Technology adoption, learning-by-doing, economies of scale, and manufacturing localization are driving the cost decrease of wind technology⁸, and similar effect could be found in the innovation and cost decrease of solar PV⁹, and storage¹⁰. Our capital costs assumptions for the Low Cost Renewable scenario for solar are a function of our estimates for the LCOE in 2030 expected given historical trends¹¹ and comparable with multiple renewable futures study.¹²⁻¹⁴ The onshore wind and battery storage capital costs are informed by the 2018 NREL Annual Technology Baseline study.¹”

References:

1. NREL. *2018 Annual Technology Baseline*. <https://data.nrel.gov/files/89/2018-ATB-data-interim-geo.xlsx> (2018).
2. NREL. *2019 Annual Technology Baseline*. <https://atb.nrel.gov/electricity/2019/files/2019-ATB-data.xlsx> (2019).
3. Jia, K. Zheng Sheng'an: 2030 Wind and Solar Capacity Will Reach 1200GW (in Chinese). *China Energy News* (2019).

4. NDRC. *China Provincial GHG Emission Inventory Guideline (in Chinese)*.
<http://www.cbcsd.org.cn/sjk/nengyuan/standard/home/20140113/download/shengjiwenshiqiti.pdf> (2011).
5. Nugent, D. & Sovacool, B. K. Assessing the lifecycle greenhouse gas emissions from solar PV and wind energy: A critical meta-survey. *Energy Policy* **65**, 229–244 (2014).
6. He, G. *et al.* SWITCH-China: A Systems Approach to Decarbonizing China’s Power System. *Environ. Sci. Technol.* **50**, 5467–5473 (2016).
7. NDRC. *Enhanced actions on climate change: China’s intended nationally determined contributions (in Chinese)*.
<http://www4.unfccc.int/submissions/INDC/Published%20Documents/China/1/China's%20INDC%20-%20on%2030%20June%202015.pdf> (2015).
8. Qiu, Y. & Anadon, L. D. The price of wind power in China during its expansion: Technology adoption, learning-by-doing, economies of scale, and manufacturing localization. *Energy Economics* **34**, 772–785 (2012).
9. Zheng, C. & Kammen, D. M. An innovation-focused roadmap for a sustainable global photovoltaic industry. *Energy Policy* **67**, 159–169 (2014).
10. Kittner, N., Lill, F. & Kammen, D. M. Energy storage deployment and innovation for the clean energy transition. *Nature Energy* **2**, nenergy2017125 (2017).
11. Amol A. Phadke, Nikit Abhyankar, Ranjit Deshmukh, Julia Szinai & Anand R. Gopal. *Cost-effective decarbonization of California’s power sector by 2030 with the aid of battery storage*. <https://ies.lbl.gov/publications/cost-effective-decarbonization> (2020).
12. NREL. *Renewable Electricity Futures Study*. <https://www.nrel.gov/analysis/re-futures.html> (2012).

13. Mai, T., Mulcahy, D., Hand, M. M. & Baldwin, S. F. Envisioning a renewable electricity future for the United States. *Energy* **65**, 374–386 (2014).
14. Williams, J. H. *et al.* The Technology Path to Deep Greenhouse Gas Emissions Cuts by 2050: The Pivotal Role of Electricity. *Science* **335**, 53–59 (2012).

REVIEWERS' COMMENTS:

Reviewer #3 (Remarks to the Author):

Dear authors, I believe that all of my points were properly addressed. The inclusion of the supplementary material and the additional explanation regarding the GHG emissions and the technical assumptions behind cost reduction curves were good enough. I believe that this paper is ready for publication.